# Tumor-Progressive Mechanisms Mediating miRNA–Protein Interaction

**DOI:** 10.3390/ijms222212303

**Published:** 2021-11-14

**Authors:** Hiroaki Konishi, Hiroki Sato, Kenji Takahashi, Mikihiro Fujiya

**Affiliations:** 1Department of Gastroenterology and Advanced Medical Sciences, Asahikawa Medical University, Midorigaoka, Asahikawa 078-8510, Japan; fjym@asahikawa-med.ac.jp; 2Gastroenterology and Endoscopy, Division of Metabolism and Biosystemic Science, Gastroenterology, and Hematology/Oncology, Department of Medicine, Asahikawa Medical University, Midorigaoka, Asahikawa 078-8510, Japan; hirokisato@asahikawa-med.ac.jp (H.S.); t-kenji@asahikawa-med.ac.jp (K.T.)

**Keywords:** microRNA, RNA-binding protein, cancer therapy

## Abstract

MicroRNAs (miRNAs) are single-stranded short-chain RNAs that are endogenously expressed in vertebrates; they are considered the fine-tuners of cellular protein expression that act by modifying mRNA translation. miRNAs control tissue development and differentiation, cell growth, and apoptosis in cancer and non-cancer cells. Aberrant regulation of miRNAs is involved in the pathogenesis of various diseases including cancer. Numerous investigations have shown that the changes in cellular miRNA expression in cancerous tissues and extracellular miRNAs enclosed in exosomes are correlated with cancer prognosis. Therefore, miRNAs can be used as cancer biomarkers and therapeutic targets for cancer in clinical applications. In the previous decade, miRNAs have been shown to regulate cellular functions by directly binding to proteins and mRNAs, thereby controlling cancer progression. This regulatory system implies that cancer-associated miRNAs can be applied as molecular-targeted therapy. This review discusses the roles of miRNA–protein systems in cancer progression and its future applications in cancer treatment.

## 1. Introduction

MicroRNAs (miRNAs), short-chain RNAs of 18–22 nt chain length, are expressed in all vertebrates and control tissue development, differentiation, cell growth, and apoptosis in non-cancer and cancer cells [1,2,3]. Additionally, miRNAs target mRNAs on the basis of their sequence and decrease protein production by inhibiting mRNA translation or destroying the mRNAs, thereby controlling cellular homeostasis. miRNA expression is controlled through DNA modification, such as by methylation and transcriptional factors through the signal-transduction pathway [4,5,6]. miRNAs are transcribed as pri-mature from DNA and are then processed by the Drosha complex, thus generating pre-miRNAs. These pre-miRNAs are transported by exportin 5 from the nucleus to the cytoplasm and further processed by Dicer to form the double-stranded miRNA RNA-induced silencing complex (RISC). RISCs involving single-strand miRNAs, comprising Ago, Dicer, and trans-activation-responsive RNA-binding protein (TRBP) 2, are directed to the mRNA targets, thus regulating protein expressions. To date, the bioinformatics databases TargetScan (http://www.targetscan.org/vert_72/, 26 September 2021) and miRTarBase (http://mirtarbase.cuhk.edu.cn/php/index.php, 26 September 2021) have been used to predict the interaction between miRNAs and target mRNAs.

The disruption of miRNA expression control is closely associated with cancer progression. For instance, miR-219-2-3p, miR-148a, and miR-335, which function as tumor suppressors by targeting oncogenic genes such as *TGIFG* and RAS p21 protein activator (GTPase-activating protein) 1, are downregulated by DNA methylation of the CpG island in gastric cancer [7,8,9,10]. Similarly, miR-342, miR-34b/c, miR-137, and miR-345 are also silenced by DNA methylation in colorectal cancer [11,12,13,14].

The tumor-progression mechanisms mediated by aberrant expressions of miRNAs are therapeutic targets for gastrointestinal cancer because some miRNAs are correlated with cancer prognosis and progression. For instance, miR-21 is a tumor-promoting miRNA because it decreases the expression of tumor-suppressive genes, including *PDCD4* and *PTEN*, and promotes the progression of gastrointestinal cancers [15,16]. Furthermore, the exosomal and cancer-tissue miR-21 are upregulated, and the detection rate of cancer-derived exosomal miR-21 is higher than that of the classical biomarkers CA19-9 and CEA in pancreatic cancer [17]. Additionally, exosomal miR-21 derived from gastric cancer cells promotes metastasis through the epithelial-to-mesenchymal transition by targeting SMAD7 mRNA in peritoneal mesothelial cells [18]. These findings support the fact that tumor-promoting miRNAs targeting tumor-suppressive mRNAs can be used as therapeutic targets for gastrointestinal cancers.

Although the translational regulation by miRNAs through the binding with mRNAs has been aggressively investigated in the previous decade, reports that focused on the direct interactions between miRNAs and proteins are few. This review summarizes the tumor-progression mechanisms mediated by the miRNA–protein interaction.

## 2. miRNA-Mediated Protein Regulation in Cancer Cells

### 2.1. RNA-Binding Proteins

In most cases, functional protein regulation by miRNAs without translational regulations through binding with mRNAs was mediated by a complex of miRNA–RNA-binding proteins (RBPs). RBPs are molecular groups with RNA-binding domains associated with RNA regulation, including the processing, transportation, splicing, translation, and stabilization of mRNAs (Figure 1A), thereby maintaining cellular functions such as differentiation, development, apoptosis, and inflammation [19,20,21]. For instance, we demonstrated the heterogeneous ribonucleoprotein (hnRNP) A1 to stabilize the trefoil factor 2, which regulates apoptosis, enhances epithelial restoration, and attenuates intestinal injury in T cell-activated enteritis model [22]. Human 424, mouse 413, fly 257, and worm 244 RBPs were registered in RBPDB (http://rbpdb.ccbr.utoronto.ca/, 28 October 2021). Additionally, 16 RNA-binding domains (RNA recognition motif [RRM], K homology [KH], CCCH zinc finger, like Sm domain, cold-shock domain [CSD], PUA domain, ribosomal protein S1-like [S1], Surp module/SWAP [SURP], Lupus La RNA-binding domain [La], PWI domain, YTH domain, THUMP domain, Pumilio-like repeat [PUM], sterile alpha motif [SAM], C_2_H_2_ zinc finger, and TROVE module), which specifically recognize the sequences, structures, or both of target RNAs, have been identified; the interaction of miRNA with RBPs are mediated by these domains. Additionally, the recognition sequences of the RNA-binding domain of each RBPs were registered in RBPDB.

Previous investigations suggested that some RBPs are directly associated with the maturation of miRNAs and transportation in cell–cell communication. hnRNP A1 directly binds to the pri-miR-18a through a specific sequence recognized by RRM of hnRNP A1 and promotes the processing and maturation of miR-18a [23]. Lin28 also directly binds to pri-let7 via CSD and CCHC zinc knuckle domain and inhibits its processing by Dicer, thereby regulating pluripotency and tissue development and differentiation [24]. Interestingly, RBPs, including AGO2, HuR, hnRNP A2/B1, YBX1, and SYNCRIP, recognize the RNA sequence motifs and/or secondary conformation and control the packaging of RNA into extracellular vesicles [25]. Therefore, this database has a high data availability and better ability to screen miRNA–RBP interactions when RBPs were not post-transcriptionally modified.

### 2.2. RBPs in Cancer Progression and Suppression

Previous investigations have suggested that RBPs are associated with the progression and suppression of various types of cancer [26]. Changes in the expression of RBPs such as hnRNP AB [27] and hnRNP K [28] have been reported in gastrointestinal cancer cells and are correlated with prognosis. RBPs exhibit cancer-promoting and suppressive functions that mediate RNA stabilization, transportation, and degradation. For instance, we showed that hnRNP H1, which is highly expressed in colorectal cancer tissues, directly binds with and stabilizes 54 apoptosis-related mRNAs, including sphingosine-1-phosphate lyase 1 mRNA, thereby promoting the growth of colorectal cancer cells [29]. Conversely, Quaking, which is downregulated in colorectal cancer, accelerates the translation of p27 and β-catenin mRNA, thereby suppressing proliferation in colorectal cancer [30]. Tristetraprolin (TTP), which is also downregulated in colorectal cancer, destabilizes VEGF mRNA and suppresses tumorigenesis in human colon cancer [31]. Interestingly, we found that some RBPs work as oncogenes without showing expressional changes [32]. In this study, we constructed 1198 siRNAs targeting human 416 RBPs and performed a functional assay based on cell growth assessment in colorectal, pancreatic, and esophageal cancer cells in vitro. Moreover, 80, 101, and 121 RBPs promoted cell growth in colorectal, pancreatic, and esophageal cancers, respectively; moreover, 41 RBPs commonly served as oncogenes in these gastrointestinal cancer cells. Among these, 12 RBPs (RPS3, RBM22, EIF2S1, DHX8, RBM8A, UPF1, YBX1, SNRPE, SF3A1, U2AF1, SUPT6H, and EIF3G) were not overexpressed in cancer cells compared with non-cancer cells, whereas 9 RBPs (DHX8, EIF3G, RBM22, SF3A1, SNRPE, SUPT6H, U2AF1, UPF1, and YBX1) did not interfere with the growth of non-cancer cells. These suggest that posttranslational modification, such as phosphorylation, SUMOylation, acetylation, and ubiquitination of RBPs, are closely associated with the functional regulation of RBPs. In fact, we previously reported that RBP modifications, such as phosphorylation, markedly influence the interactions among RNAs and RBPs. Our study demonstrated that the phosphorylation status of hnRNP A0 was augmented in colorectal cancer tissues compared with normal tissues, and this enhanced cancer progression by binding to and stabilizing RAB3GAP1 mRNA in colorectal cancer cells [33]. The knockdown of hnRNP A0 induced apoptosis in colorectal cancer cells (except in non-cancer epithelial cells) because of minimal binding between hnRNP A0 and RAB3GAP1 mRNA in non-cancerous epithelial cells. Similarly, Yuan et al. demonstrated that phosphorylated KHSRP, which recognizes single-stranded nucleic acids using its four KH domains and specifically short G-rich stretches in the terminal loop (TL-G-rich) of primary or precursor miRNAs, promotes the maturation of let7 by the phosphorylation of its Ser193 residue; however, it also inhibits its maturation by SUMOylation of K87 residue; thus, SUMOylated KHSRP is associated with tumorigenesis [34]. Furthermore, Li et al. demonstrated that SUMOylated hnRNP A1 supports the leading tumor-promoting miRNAs in extracellular vesicles and exosomes, thereby inducing tumor proliferation and migration in non-small cell lung cancer cells [35]. Therefore, post-transcriptional cancer-specific modifications, such as phosphorylation and SUMOylation, were associated with the interactions between RBPs and RNAs, including mRNAs and miRNAs. Establishing novel databases, including those for target RNAs by cancer-specific and post-translationally modified RBPs, will unveil the RBP–RNA interaction networks in cancer cells and allow the identification of attractive therapeutic targets that have minor adverse effects.

### 2.3. miRNA–RBP Binding Functions in Cancer Cells

Although 2654 miRNAs (*Homo sapiens*) have been registered in the database (http://miRbase.org/, 28 October 2021), only a few miRNAs that bind to target proteins and change the cellular functions without a direct binding between miRNA and mRNA have been identified (Table 1). We searched the PubMed database using the keywords “(miRNA) AND (decoy) NOT (circRNA) NOT (lncRNA) AND (cancer)” and found 53 articles. Of these, we selected original studies on miRNA-mediated RBP functional regulation in cancer progression or suppression.

The most-investigated function of direct binding between miRNA and RBPs is the “decoy” system (Figure 1B,C). miRNA directly binds to target RBPs through its RNA-binding domain (RBD) on the basis of their sequences and cancels out the functions of RBP, including the inhibition/promotion of mRNA translations. This system was first reported by Eiring et al. in leukemic blast cells. They showed that miRNA-328 bound to hnRNP E2, releasing CCAAT/enhancer-binding protein alpha (C/EBPα) mRNA from hnRNP E2, thereby restricting the translation of C/EBPα mRNA and supporting the differentiation of progenitor cells in leukemic blast cells (Figure 2A) [36]. Interestingly, miRNA-328 interacts with hnRNP E2 without RISC-associated proteins, such as Ago, suggesting that this system works independently of the mRNA-associated gene silencing mechanism of miRNA. In contrast, Balkhi et al. revealed that miR-29, which has a complementary sequence of 3′ UTR of tumor-suppressive TNFAIP3 mRNA, was decreased in patients with sarcoma, and it directly bound to the RBPs HuR, inhibited the recruitment of RISC to the 3′ UTR of TNFAIP3 mRNA, and negatively regulated NFkB signaling, thereby suppressing tumorigenesis in sarcoma cells [37]. We also globally assessed hnRNP A1-binding RNAs through microarray and whole transcriptome analyses combined with RNA immunoprecipitation. The results demonstrated that miR-26a, miR-584, and CDK6 mRNA had a high affinity for the RBD of hnRNP A1. The induction of miR-26a or miR-584 inhibited the binding between hnRNP A1 and CDK6 mRNA, which is recognized and stabilized by hnRNP A1 mediating RBD, and decreased CDK6 expression, resulting in apoptosis induction in colorectal cancer cells (Figure 2B) [38]. Yao et al. revealed that miR-574-3p works as a decoy for hnRNP L, which supports the translation of VEGFA mRNA by interacting 3′ UTR-localized CA-rich elements and inhibiting the translation of VEGF, resulting in tumor suppression in lymphoma cells [39]. Saul et al. revealed that miR-574-5p, which contains a GU-rich sequence, is highly induced in patients with non-small cell lung cancer and acts as an RNA decoy to CUG RNA-binding protein 1 (CUGBP1), which suppress the translation of microsomal prostaglandin E synthase-1 (mPGES-1) by directly binding mediating CU-rich element 1 and 2 in 3′ UTR of mPGES-1 mRNA, and antagonizes the function of CUGBPs, thereby supporting tumorigenesis in non-small cell lung cancer [40].

Additionally, we demonstrated that miRNA-18a, which is overexpressed in colorectal cancer cells, is directly bound with hnRNP A1 and induces the ubiquitin-autophagosomal degradation of hnRNP A1 through direct binding with RBD of hnRNP A1, leading to cancer cell apoptosis (Figure 1D). Interestingly, this miRNA-18a function was inhibited by a short RNA sequence recognized by the RRM of hnRNP A1 [41]. Notably, this post-transcriptional modification system brings about the degradation of target proteins without interfering with RBP–mRNA binding. Thus, these reports show that miRNA–RBP binding has at least two functions, including controlling mRNA transcription by inhibiting the RBP–mRNA binding, such as through a decoy, and the degradations of RBPs induced by the direct binding between RBPs and miRNAs.

### 2.4. Strategies for Identifying Interactions between miRNAs and RBPs

Most reports have analyzed and identified specific miRNA and RBP decoy systems using expressional analysis targeting miRNAs and complementary sequences recognized by the RBD of RBPs. For instance, Eiring et al. compared the miRNA expressions in patient-derived chronic myelocytic leukemia (CML)-blast crisis (BC) CD34+ versus CML-chronic phase (CP) CD34+ bone marrow progenitors using microarray analysis and found that various miRNAs, including miRNA-328, were downregulated in CML-BC. They focused on miR-328 because its mature form harbors a C-rich sequence that resembles the negative regulatory hnRNP E2-binding site included in the CEBPA intercistronic mRNA region [36]. However, previous studies, including our study, have shown that the binding affinities of RBPs and RNAs are influenced by the posttranslational modification of RBPs. Therefore, the direct binding must be confirmed through other molecular biological methods, such as pulldown assays combined with RT-PCR and electrophoretic mobility shift assays. Notably, specific RBP–miRNA interactions based on the RBD of RBP cannot rank the affinity of each miRNA–RBP binding in cellular physiological conditions.

A useful strategy for identifying novel miRNA–RBP interactions is RNA immunoprecipitation (RNA-IP) combined with microarray analysis [38] (Figure 3). In this study, the cancer-associated RBP hnRNP A1 was subjected to a pulldown assay using immunoprecipitation in colorectal cancer cells. RNAs were eluted from the precipitant via phenol–chloroform extraction, and microarray analysis was conducted to detect the miRNAs interacting with RBPs. Notably, this method determines the physical binding of miRNAs and RBPs in cancerous cells; therefore, the tumor therapeutic binding of miRNA and RBP is enhanced. As listed in Table 2, numerous miRNAs are directly bound to hnRNP A1. Furthermore, RBPs have a high affinity for a specific RNA motif recognized by the RBD. To confirm the RBP–miRNA binding, RNA competencies of miRNAs, which are chemically synthesized artificial short RNA similar to the RNA-binding motif sequence, were developed. We demonstrated that RNA competence could inhibit the specific binding of miR-26a, miR-584, and hnRNP A1, thus preventing miRNA-induced apoptosis in colorectal cancer cells [38]. In another study, specific RNA competence against the binding between hnRNP A1 and miR-18a also inhibited the degradation of hnRNP A1 by the miRNA; this thereby canceled the growth suppression due to miR-18a overexpression [41]. These results demonstrate that the length of RNA competence that binds to RBPs is one of the important factors in controlling the RBP–RNA competition system and RBP degradation system mediating the binding of RNA competence. By understanding the specific mechanisms underlying RBP–RNA binding and the length of the RNA competitor, we can choose from among dual cancer therapeutic strategies, i.e., whether to destroy the target protein to eliminate the tumor-promoting function completely or to obtain strong therapeutic effects and inhibit RBP functions temporally to reduce adverse effects.

Additionally, miRNA and RBP docking simulation based on scientific calculation techniques will support the identification of novel miRNA–RBP interactions. This technique can simulate the affinity of proteins and ligands, such as nucleotides, based on the protein structures deposited in the Protein Data Bank and conformation of ligands. For example, we previously demonstrated that the affinity of hnRNP A0 and nucleotides is modified by the phosphorylation status of hnRNP A0 [33]. Furthermore, the affinity of tumor-specific modified RBPs and miRNAs can be estimated using this method. This technique will therefore enable an understanding of miRNA–RBP binding in cancer progression.

## 3. Future Perspectives

Previous studies have demonstrated that some RBPs have strong tumor-promoting functions and thus can be used as a cancer therapeutic target. However, RBPs have essential functions in non-cancerous cells, such as tissue formation and repair [42]. Thus, adverse effects are worrisome when RBPs are applied in cancer treatment. Notably, we identified 12 RBPs that have tumor-promoting functions with less expressional changes at the mRNA level [32]. Similarly, the tumor-promoting functions of certain RBPs, including hnRNP A0 and KHSRP, depend on their posttranslational modification, including phosphorylation and SUMOylation; thus, tumor-specific modification of RBPs may be a useful cancer therapeutic target. An antibody-mediated therapeutic strategy, a target for posttranslational modification, is now under development. However, nearly all RBPs are present in the cytoplasm or nucleus. To achieve therapeutic efficacy via antibody-based therapy, modification of the drug-delivery system of antibodies is needed to allow them to pass through the plasma and/or nuclear membrane. Therefore, RBP-targeted strategies, which are convenient as well as effective, are warranted to achieve cancer therapeutic effects.

As described in this review, the use of miRNA in cancer therapeutics and as diagnostic markers is based on its expressional changes in cancer cells and cancer cell-derived extracellular vesicles. Additionally, miRNAs target mRNA and inhibit the translation and cellular physiological functions, including cell growth, development, and apoptosis. Dysregulation of this function leads to cancer progression. Notably, this review highlights miRNAs can also target the RBPs and inhibit their oncogenic/tumor-suppressive functions. Thus, this non-canonical system can be used as a novel therapeutic strategy for various cancers, including gastrointestinal cancers.

Cancer therapeutic strategies using miRNA–RBP interactions have some advantages. First, there are highly flexible through the artificial modification of the miRNA sequences. By modifying the miRNA sequence, the affinity of miRNA–RBP can be regulated in each tumor type, and unexpected targets, including mRNAs, will be controlled. Moreover, the adverse effects observed after miRNA administration to patients with cancer can be regulated by administering siRNAs corresponding to the miRNA. Second, RNA competence modified on the basis of the specific target miRNA can specifically inhibit the RBP–miRNA binding, thus controlling RBP-associated mRNA stability and RBP degradation. Third, because RNA competencies are short, a pivotal RISC component, Ago1, cannot bind the RNA competences. Ago1 binds with the miRNA duplex by recognizing the mismatch of the miRNA duplex in the central region (guide position 9–11) and accelerates the maturation of Ago1-RISC [43]. This suggests that RNA competencies directly inhibit the RBP–miRNA interaction without affecting the endogenous miRNA–mRNA interaction mediating RISC. This further indicates that RNA competence therapy as well as miRNA therapy is expected to lead to few adverse events while RBPs control numerous cellular functions.

Meanwhile, certain assignments should be resolved using miRNA–RBP interactions for cancer treatment. First, an effective system to deliver miRNAs to cancer cells has not yet been established. For this, developing oligonucleotide therapeutics has been interrupted. Recently, extracellular vesicle and nanocarrier systems for nucleotides are being developed to combat refractory cancer treatment [44,45]. These approaches will resolve the drug-delivery problem in nucleotide therapeutics, including miRNAs. Second, the instability of nucleotide therapeutics in vivo is one of the major assignments. Furthermore, a large amount of RNA must be administered to patients to achieve the desired therapeutic effects in refractory cancer using RNA drugs. However, the mammalian body has a defensive system for exogenous nucleic acids. For instance, TLR3 recognizes the double-strand RNA of the virus, TLR7 and 8 recognize the single-strand RNA of the virus, and TLR9 distinguishes DNA from the bacteria or virus [46]. Therefore, adverse events may occur when abundant nucleic acid is administered to the patient. The clinical phase 3 study of the RNA aptamer REG1 has been discontinued as there were severe allergic reactions. To decrease the number of nucleic acid drugs to avoid these adverse events, locked nucleic acid, which is artificially generated nucleic acid intended to increase the resistance for ribonuclease, has been highlighted. This technology will strongly promote the development of nucleic acid drugs. Third, miRNA has numerous mRNA targets and RBPs; thus, unexpected reactions can occur. As mentioned above, the sequence of nucleotide therapeutics can be modified to reduce these adverse effects, whereas novel targets could be targeted, causing unexpected adverse effects. Administering a minimum miRNA sequence, such as RNA competence, which interacts with specific RBPs, may reduce non-specific mRNA binding.

As discussed in this review, the miRNA–protein interaction is an attractive therapeutic target for gastrointestinal cancer. However, investigations focused on the miRNA–protein interaction are far less in number than those on the miRNA–mRNA interaction. Unfortunately, the therapeutic candidate miRNA–RBP binding has been less identified. We propose an original identification method involving a combined analysis of tumor-specific binding between miRNA and RBP and comprehensive analyses. Involving various fields, such as medical biology, chemistry, and physics, will help determine the unknown therapeutic targets mediating the miRNA–RBP interaction in the future.

## Figures and Tables

**Figure 1 ijms-22-12303-f001:**
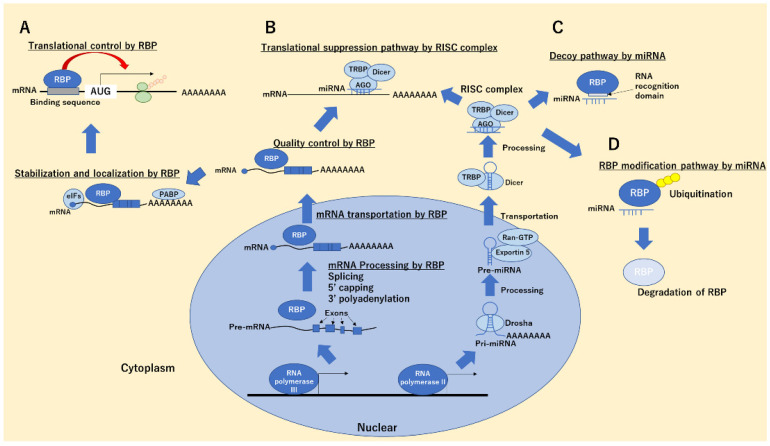
Working system of microRNA (miRNA) targeting RNA-binding protein (RBP). RBPs support the regulation of translation by binding to messenger RNAs (mRNAs) (**A**). Conversely, microRNA (miRNA)-containing RNA-induced silencing complexes (RISCs) suppress the translation through the sequence-dependent binding to mRNAs (**B**). Some miRNAs, which have similar sequences with RNA-binding domains of RBP, can be decoy mRNA and interfere with the translation mediating the RBP–mRNA binding (**C**) and induce RBP ubiquitination mediating the direct binding (**D**).

**Figure 2 ijms-22-12303-f002:**
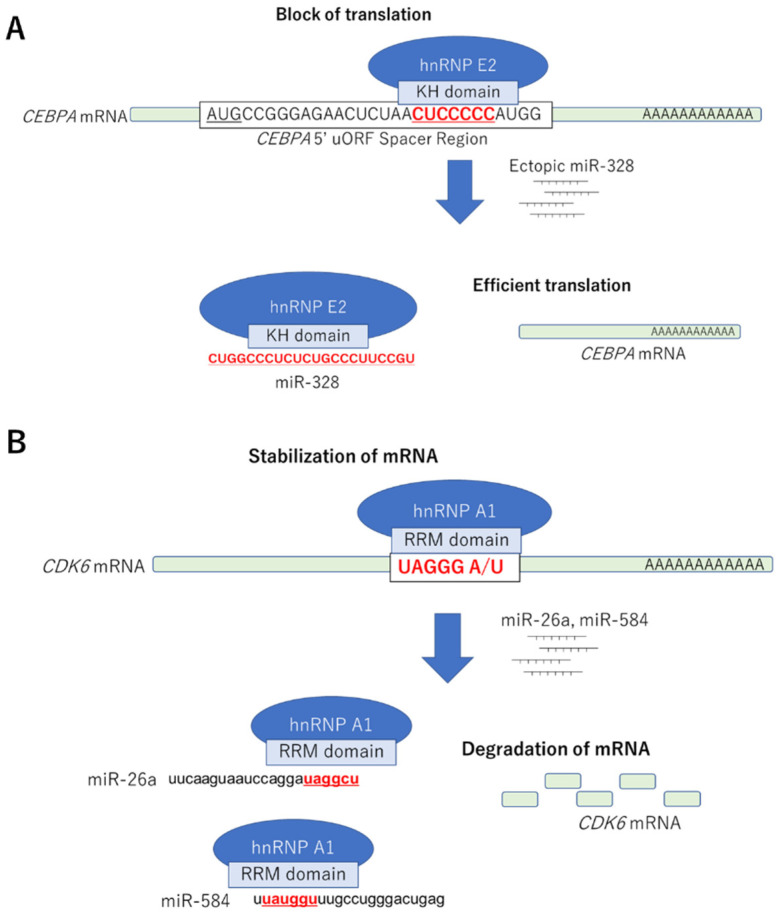
Decoy system mediating miRNA. miR-328 directly binds to the KH domain of hnRNP E2 and inhibits the binding of hnRNP E2 and CEBPA mRNA, thereby accelerating the translation of CEBPA mRNA (**A**). miR-26a and 584 directly bind to the RRM domain of hnRNP A1 and inhibit the binding of hnRNP A1 and CDK6 mRNA, thereby inducing the destabilization of CDK6 mRNA (**B**).

**Figure 3 ijms-22-12303-f003:**
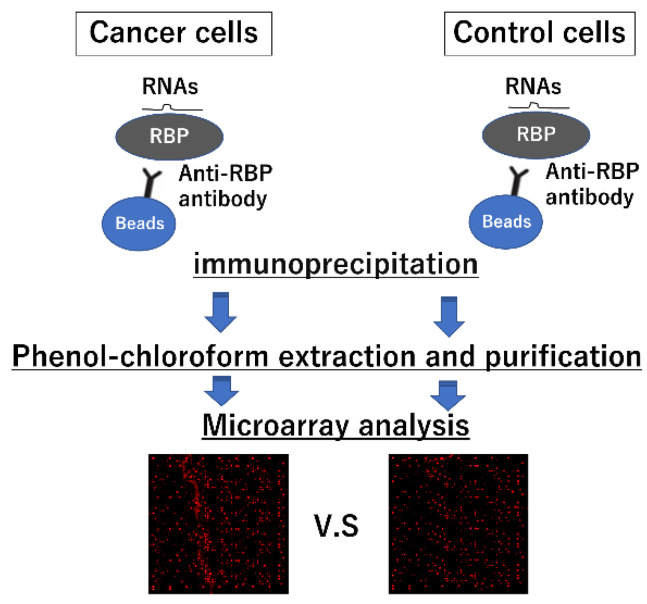
Methodology for the identification of RNA-binding protein (RBP)-binding micro RNAs (miRNAs) and messenger RNAs (mRNAs). RNA-IP with microarray analysis is a powerful strategy to exhaustively identify unknown PBP-miRNA or messenger RNA (mRNA) interactions. miRNAs and mRNAs, which interact with a specific RNA-binding protein (RBP) in cancer cells, are subjected to pulldown assays by immunoprecipitation using anti-RBP antibodies extracted through the phenol–chloroform extraction method and identified via microarray analysis.

**Table 1 ijms-22-12303-t001:** Protein-targeted miRNAs and regulation systems.

miRNA	Type of Pathway	Target	Type of Cancer	Function	Reference
miR-328	Decoy	hnRNP E2-CEBPα mRNA	Leukemic blasts	Differentiation	Eiring AM, Cell 2010
	Canonical	PIM1 mRNA		Decreased survival	
miR-29	Decoy	HuR-A20 mRNA	Sarcoma	Differentiation	Balkhi MY, Sci signal, 2013
miR-26a, -584	Decoy	hnRNP A1-CDK6 mRNA	Colorectal cancer	Cell growth suppression	Konishi H, Biochem Biophys Res Commun. 2015
miR-574-3p	Decoy	hnRNP L-VEGFA mRNA	Myeloid cells	Inhibition of cell proliferation	Yao P, Nucleic Acids Research, 2017
	Canonical	EP300 mRNA			
miR-574-5p	Decoy	CUGBP1-mPGES-1 mRNA	Lung tumor	Cell growth promotion	Saul MJ, FASEB J, 2019
miR-18a	Degradation	hnRNP A1	Colorectal cancer	Apoptosis induction	Fujiya M, Oncogene, 2014

**Table 2 ijms-22-12303-t002:** hnRNP A1 interacting miRNAs identified by RNA-IP in combination with microarray analysis.

miRs with Greater than 4-Fold Expression Compared to the Isotype Control IgG			
Name	ID	Ratio (hnRNP A1/IgG)	LOG2ratio
hsa-miR-29a-3p	MIMAT0000086	11.49	3.52
hsa-miR-26a-5p	MIMAT0000082	11.37	3.51
hsa-miR-584-5p	MIMAT0003249	9.93	3.31
hsa-miR-107	MIMAT0000104	9.73	3.28
hsa-miR-106b-5p	MIMAT0000680	8.99	3.17
hsa-miR-1229-5p	MIMAT0022942	8.88	3.15
hsa-miR-29b-3p	MIMAT0000100	8.07	3.01
hsa-miR-194-5p	MIMAT0000460	8.07	3.01
hsa-miR-142-3p	MIMAT0000434	7.97	2.99
hsa-miR-18a-5p	MIMAT0000072	7.93	2.99
hsa-let-7c-5p	MIMAT0000064	7.31	2.87
hsa-miR-16-5p	MIMAT0000069	7.22	2.85
hsa-miR-500a-3p	MIMAT0002871	6.96	2.8
hsa-miR-200b-3p	MIMAT0000318	6.89	2.78
hsa-miR-19a-3p	MIMAT0000073	6.55	2.71
hsa-miR-222-3p	MIMAT0000279	6.4	2.68
hsa-let-7b-5p	MIMAT0000063	5.98	2.58
hsa-miR-23a-3p	MIMAT0000078	5.97	2.58
hsa-let-7d-5p	MIMAT0000065	5.85	2.55
hsa-miR-431-3p	MIMAT0004757	5.7	2.51
hsa-miR-200c-3p	MIMAT0000617	5.62	2.49
hsa-miR-23b-3p	MIMAT0000418	5.27	2.4
hsa-miR-27b-3p	MIMAT0000419	5.23	2.39
hsa-miR-19b-3p	MIMAT0000074	5.05	2.34
hsa-miR-103a-3p	MIMAT0000101	4.95	2.31
hsa-miR-1246	MIMAT0005898	4.73	2.24
hsa-let-7a-5p	MIMAT0000062	4.66	2.22
hsa-miR-20a-5p	MIMAT0000075	4.5	2.17
hsa-miR-27a-3p	MIMAT0000084	4.49	2.17
hsa-miR-141-3p	MIMAT0000432	4.32	2.11
hsa-miR-21-5p	MIMAT0000076	4.25	2.09
hsa-miR-17-5p	MIMAT0000070	4.24	2.08
hsa-miR-106a-5p	MIMAT0000103	4.15	2.05
hsa-miR-20b-5p	MIMAT0001413	4.11	2.04

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
