# Peer review of "Tumor-Progressive Mechanisms Mediating miRNA–Protein Interaction"

_ijms, 2021, doi:10.3390/ijms222212303_

Round 1
Reviewer 1 Report
While the translational regulation through miRNAs direct binding to mRNAs has been intensively studied, interactions between miRNAs and proteins are underenvestigated. This review is focused on miRNA-protein interactions in cancer.
Authors should decide whether they discuss miRNA-protein interactions in total or particularly in gastrointestinal cancers. In the first case, gastrointestinal cancers should be excluded from the abstract, in the second case, – included in the title. Maybe this shift of the emphasis should be reflected in the main text too in some cases.
In some paragraphs, authors widely discuss their own data obtained previously and in total cite 8 of their own works. Are there no other data on the topic available? If so, it should be mentioned. How the literature search was carried out?
In the paragraph 2.2 authors emphasize that RBPs exhibit cancer-promoting functions (page. 2, line 96) and further discuss these functions. It is a very one-sided view, since RBPs have much more complex functions including those involved in cancer suppression (see review DOI: 10.3390/cancers12092699 for example).
Page 3, line 135. Since the majority of 38,589 miRNAs indexed in miRBase are actually homologs from different species, I suggest using the number of miRNAs known for one specific species, preferably for H. sapiens (2654) instead 38,589
Author Response
While the translational regulation through miRNAs direct binding to mRNAs has been intensively studied, interactions between miRNAs and proteins are underenvestigated. This review is focused on miRNA-protein interactions in cancer.
Authors should decide whether they discuss miRNA-protein interactions in total or particularly in gastrointestinal cancers. In the first case, gastrointestinal cancers should be excluded from the abstract, in the second case, – included in the title. Maybe this shift of the emphasis should be reflected in the main text too in some cases.
Response: Thank you for pointing this out. We eliminated the word “gastrointestinal” from the abstract.
In some paragraphs, authors widely discuss their own data obtained previously and in total cite 8 of their own works. Are there no other data on the topic available? If so, it should be mentioned. How the literature search was carried out?
Response: As the reviewer specified, there are few reports on protein regulation systems mediating miRNA. We searched Pubmed using as keyword (miRNA) AND (decoy) NOT (circRNA) NOT (lncRNA) AND (cancer), and 53 articles were found. Of these, we selected original papers on the miRNA-mediated RBP functional regulation in cancer progression or suppression.
In the paragraph 2.2 authors emphasize that RBPs exhibit cancer-promoting functions (page. 2, line 96) and further discuss these functions. It is a very one-sided view, since RBPs have much more complex functions including those involved in cancer suppression (see review DOI: 10.3390/cancers12092699 for example).
Response: Thank you for this suggestion. We discussed the tumor suppressive functions of some RBPs in this section.
Page 3, line 135. Since the majority of 38,589 miRNAs indexed in miRBase are actually homologs from different species, I suggest using the number of miRNAs known for one specific species, preferably for H. sapiens (2654) instead 38,589
Response: Thank you for this suggestion. We changed the number of miRNAs in H. sapiens as suggested.
Reviewer 2 Report
The review of Konishi et al. is well written and can be published in the present form.
Author Response
The review of Konishi et al. is well written and can be published in the present form.
Response; Thank you for review of our manuscript.
Reviewer 3 Report
While the translational regulation by miRNAs through binding with mRNAs has been aggressively investigated, reports focused on direct interactions
between miRNAs and proteins are far less studied . This review summarizes the tumor-progressive mechanisms mediated by miRNA–protein interaction. In this review the authors have covered a new area which has not been thoroughly studied in the past. Unfortunately however , the review is poorly written and need extensive improvement before it might be considered for publication. The reviewer suggests incorporating sequences of specific miRNAs and details of how they interact with RNAs in the decoy pathway in Figs like 1 and 2 as examples. If possible they can do with more than one or two miRNAs This will clearly highlight how these interactions differ from more established interactions between miRNAs and its target RNAs through seed sequence of miRNAs and 3'UTRs of target mRNAs. In the absence of such information it may be difficult to evaluate the importance of such pathway.
Author Response
While the translational regulation by miRNAs through binding with mRNAs has been aggressively investigated, reports focused on direct interactions between miRNAs and proteins are far less studied. This review summarizes the tumor-progressive mechanisms mediated by miRNA–protein interaction. In this review the authors have covered a new area which has not been thoroughly studied in the past. Unfortunately however, the review is poorly written and need extensive improvement before it might be considered for publication. The reviewer suggests incorporating sequences of specific miRNAs and details of how they interact with RNAs in the decoy pathway in Figs like 1 and 2 as examples. If possible they can do with more than one or two miRNAs This will clearly highlight how these interactions differ from more established interactions between miRNAs and its target RNAs through seed sequence of miRNAs and 3'UTRs of target mRNAs. In the absence of such information it may be difficult to evaluate the importance of such pathway.
Response: Thank you for your suggestion. We agree with your comment and have added the new figure that shows the decoy system of miR-328, -26a, and -584. miR-328 is the first miRNA identified, which works as a decoy for RBP and hnRNP E2. hnRNP E2 suppresses the translation of CEBPA mRNA by directly binding to the C-rich sequence of mRNA that mediates the KH domain. miR-328 also has a C-rich sequence and alternatively binds to the KH domain of hnRNP E2, thereby annulling the suppression of CEBPA mRNA translation by hnRNP E2. We previously identified miR-26a and -584 by IP-microarray analysis. These miRNAs have a binding motif for the RRM domain of hnRNP A1. CDK6 mRNA also has a binding motif for the RRM domain, and IP-transcriptome analysis showed the direct binding of hnRNP A1 and CDK6 mRNA in SW620 cells. Overexpression of miR-26a or -584 occupies the RRM domain of hnRNP A1 and annuls the binding between CDK6 mRNA and hnRNP A1, leading to the instability of CDK6 mRNA. This figure and legends have been added in Figure 2.
Round 2
Reviewer 1 Report
I suggest authors to perform additional check of the writing style and improve it where it is possibble.
Author Response
I suggest authors to perform additional check of the writing style and improve it where it is possibble.
Response; Native speaker has checked and corrected English style.
Reviewer 3 Report
The reviewer is satisfied with revisions and the manuscript may be suitable for publication
Author Response
The reviewer is satisfied with revisions and the manuscript may be suitable for publication
Response; Thank you for reviewing.
